# Relation Between Inflammatory Parameters and Insulin Resistance Indices in Cows During Early Lactation

**DOI:** 10.3390/metabo15110751

**Published:** 2025-11-20

**Authors:** Marko Cincović, Dragica Stojanović, Radojica Djoković, Mira Majkić, Jože Starič, Miloš Petrović, Zorana Kovačević

**Affiliations:** 1Faculty of Agriculture, University of Novi Sad, Square Dositeja Obradovića 7, 21000 Novi Sad, Serbia; dragicas@polj.edu.rs (D.S.); miramajkic@gmail.com (M.M.); zorana.kovacevic@polj.edu.rs (Z.K.); 2Faculty of Agronomy, University of Kragujevac, Cara Dušana 34, 32000 Čačak, Serbia; radojicadjokovic@gmail.com (R.D.); petrovic.milos87@yahoo.com (M.P.); 3Veterinary Faculty, University of Ljubljana, Gerbičeva 60, 1000 Ljubljana, Slovenia; joze.staric@vf.uni-lj.si

**Keywords:** lipolysis, insulin sensitivity, inflammation, nonsteroidal anti-inflammatory drug

## Abstract

**Background/Objectives**: Early lactation in high-producing dairy cows is a critical period characterized by pronounced negative energy balance, enhanced lipomobilization, and the development of insulin resistance (IR), often accompanied by low-grade systemic inflammation. This study aimed to investigate the dynamics of inflammatory markers and IR indices in early-lactation cows, assess their interrelationships, and evaluate the effects of NSAID administration. **Methods**: Thirty Holstein–Friesian cows were included and allocated into a control group (*n* = 15) and a treatment group (*n* = 15), which received ketoprofen (3 mg/kg BW intramuscularly) during the first postpartal week. Blood samples were collected at weeks 0, 1, and 2 postpartum to measure TNF-α, IL-1β, haptoglobin, fibrinogen, NEFA, glucose, and insulin concentrations. Surrogate indices of IR, including RQUICKI, HOMA-IR, QUICKI, and Adipo-IR, were calculated. **Results**: In the control group, TNF-α, IL-1β, haptoglobin, fibrinogen, and NEFA progressively increased over the first two weeks, accompanied by elevated adipose tissue IR, evidenced by decreased RQUICKI and increased Adipo-IR. Positive correlations were observed between inflammatory markers and NEFA, as well as between TNF-α and IL-1β with Adipo-IR. Conversely, negative correlations were found between inflammatory markers and glucose and insulin, and between TNF-α and RQUICKI, as well as IL-1β and haptoglobin with glucose. **Conclusions**: Ketoprofen administration significantly reduced inflammatory markers and NEFA while improving RQUICKI and Adipo-IR, without altering the overall relationships among the parameters. These findings indicate that inflammatory cytokines and adipose tissue IR indices serve as reliable parameters for monitoring the interaction between inflammation and IR, and for assessing the metabolic effects of NSAID treatment in early-lactation cows.

## 1. Introduction

Early lactation represents the most critical period in the production cycle of high-yielding dairy cows, characterized by substantial metabolic challenges and exceptionally high energy demands [1]. The onset of lactation inevitably leads to a state of Negative Energy Balance (NEB), in which energy intake is insufficient to meet the requirements for milk production and basal metabolism. The intensity and duration of NEB directly influence the metabolic adaptation, productivity, and overall health of the cows [2]. In response to NEB, cows mobilize body fat reserves, resulting in elevated concentrations of non-esterified fatty acids (NEFA) in the blood. The increased NEFA concentrations, together with the redirection of glucose toward the mammary gland, are key factors contributing to the development of physiological, and often pathological, insulin resistance (IR) [3].

IR, defined as a reduced ability of target tissues (muscle, liver, adipose tissue) to effectively respond to circulating insulin, represents an essential adaptation during early lactation, allowing for glucose sparing and its redirection toward the mammary gland [3]. However, when this resistance becomes excessive, it is closely associated with metabolic disorders such as subclinical and clinical ketosis, fatty liver, and altered immune function [4,5]. Insulin resistance in adipose tissue during early lactation is physiological, facilitating the mobilization of fat to meet the energetic demands of lactation; however, its regulation is crucial, as excessive adipose tissue insulin resistance can contribute to systemic insulin resistance affecting muscle and liver, potentially leading to metabolic disorders such as ketosis and fatty liver. In ruminant studies, validated surrogate indices based on basal glucose and insulin concentrations are commonly used, providing reliable insights into metabolic status [6,7]. Indices such as the Quantitative Insulin Sensitivity Check Index (QUICKI) and the Homeostasis Model Assessment of IR (HOMA-IR) are widely accepted for evaluating systemic insulin sensitivity [8]. Additionally, the Adipose tissue Insulin Resistance (Adipo-IR) index, which incorporates the relationship between insulin and NEFA, as well as the Revised QUICKI (RQUICKI), which extends the QUICKI index by including NEFA values, offer specific assessments of adipose tissue IR, which is particularly relevant under conditions of intensive lipomobilization during early lactation [9].

An increasing body of evidence suggests that IR is not merely a disorder of carbohydrate and lipid metabolism, but is deeply intertwined with a state of chronic, low-grade systemic inflammation, often referred to as “meta-inflammation” [10]. Metabolic stress, induced by elevated NEFA concentrations and other factors, activates the innate immune system and leads to increased production of pro-inflammatory cytokines [11,12]. These cytokines directly interfere with insulin signaling pathways, thereby exacerbating IR and creating a vicious cycle [13].

In this context, key inflammatory mediators of interest include the pro-inflammatory cytokines Tumor Necrosis Factor-alpha (TNF-α), Interleukin 1 beta (IL-1β), and others. These cytokines are known to reduce GLUT4 transporter translocation and inhibit insulin receptor phosphorylation, leading to alterations in insulin sensitivity [14]. Furthermore, cytokines may be associated with adipose tissue inflammation and contribute to increased lipolysis [15]. In addition to cytokines, reliable markers of systemic stress and inflammation are acute-phase proteins (APPs). Examples are haptoglobin (Hp) and fibrinogen, whose concentrations increase significantly in response to subclinical or clinical inflammatory events, serving as robust indicators of general health status and metabolic stress response during early lactation [16,17].

Given the strong link between inflammation and IR, there is considerable interest in therapeutic strategies aimed at alleviating inflammatory burden. Ketoprofen, a Non-Steroidal Anti-Inflammatory Drug (NSAID), exerts its effects through non-selective inhibition of cyclooxygenases (COX-1 and COX-2), thereby effectively reducing the synthesis of pro-inflammatory prostaglandins [18]. Although ketoprofen use has traditionally been directed toward the treatment of clinical conditions (e.g., metritis, mastitis, digital dermatitis), an increasing number of studies suggest the potential benefit of NSAIDs in modulating metabolic status by reducing systemic inflammation in high-yielding dairy cows [19,20]. Non-selective inhibition of cyclooxygenases, lipoxygenase inhibition, rapid onset of action, and short milk withdrawal make ketoprofen suitable for this longitudinal study.

It is hypothesized that reducing inflammation may improve insulin signaling and thereby mitigate the pathological aspects of IR. Although the effects of ketoprofen administration on individual metabolic or inflammatory markers have been partially investigated [21], there is a lack of comprehensive studies that simultaneously integrate a broader panel of IR indices and inflammatory markers in the context of ketoprofen use in lactating cows during early lactation. The aim of this study was to examine the effects of ketoprofen administration on inflammatory parameters and indicators of IR, as well as to thoroughly evaluate the relationship between surrogate indices of IR (HOMA-IR, QUICKI, RQUICKI, Adipo-IR) and a panel of key inflammatory markers, including cytokines (TNF-α, IL-1β) and acute-phase proteins (haptoglobin, fibrinogen). We hypothesize that ketoprofen administration will significantly alter the concentrations of inflammatory markers, carbohydrate and lipid metabolism indicators, and IR indices, and that the correlations will reflect the interplay between inflammation and IR.

## 2. Materials and Methods

### 2.1. Animals and Management

A total of 30 clinically healthy cows of Holstein–Friesian bread on a commercial dairy farm were included in the experiment. The cows were in their second or third lactation, exhibited a body condition score typical for the postpartum period (BCS 2.75–3.25, score system 1–5), and had produced 9000–10,000 L of milk in the previous lactation. The cows were divided into two groups: 15 cows treated with ketoprofen and 15 cows in the negative control group. Experimental cows were treated during first week of lactation with ketoprofen with 3 mg × kg^−1^ BW of ketoprofen IM injection (Mediprofen^®^, Vetmedic, Vršac, Serbia).

Cows were kept in standard free stalls system of cow housing. Water was available *ad libitum*. Cows were fed twice daily using the TMR mixture according to standards [22]. Components of meal (in kg of dry matter) included: corn silage-multiple grains 8.2; Haylage of alfalfa 1.65; Hay of alfalfa 1.94; ensiled beet pulp 1.8; fresh beer trub 0.96; corn grain 5.38; barley grain 0.61; rapeseed meal 0.92; soybean meal 44% 1.1; sunflower meal 33% 1.44; extruded flaxseed 0.55; livestock chalk 0.13; livestock salt 0.06; baking soda 0.06; MgO 0.05; premix 0.18; Phosphozel 0.12; Zenural (urea) 0.18; Bentonite (Mycotoxin adsorbent) 0.02; dairyfat c16 0.39.

### 2.2. Blood Sampling and Metabolic Parameters Analysis

Blood samples were collected in the 0, 1 and 2 weeks (0, 7 and 14 days after calving, respectively) around calving by venipuncture of *v.coccigea* using 10 mL serum separation tubes. In order to separate the serum better, it was additionally centrifuged for 5 min at 3000 g. The serum samples were then collected and placed in vials and transported by laboratory refrigerators to the Laboratory of Pathophysiology, Department of Veterinary Medicine, University of Novi Sad.

The concentrations of the TNF-α, IL-1β, Haptoglobin and Fibrinogen inflammatory parameters were measured by standard ELISA kit manufactured by Cloud-Clone Corp (China, Intra-Assay: CV < 10%; Inter-Assay: CV < 12%). The following readers were used for measurements: Fluoroscan Ascent FL reader (Thermo Scientific, USA) and Rayto RT 2100C reader (Rayto Life and Analytical Sciences, China).

We performed analysis for the following endocrine–biochemical parameters: insulin, non-esterified fatty acids (NEFA), glucose (GLU). Standard kits from Randox (UK) for NEFA and BioSystem (Spain) for glucose were used on Rayto Chemray 120 biochemical analyzer (Rayto Life and Analytical Sciences, China). An automated immunoassay analyzer TOSOH AIA-360 (Tosoh Bioscience, Tokyo, Japan) was used for insulin analyses. For the estimation of IR we used RQUICKI, Adipo-IR, HOMA-IR and QUICKI index according to standard formula [6,23]:

*RQUICKI = 1/[log(Glucose) + log(Insulin) + log(NEFA)]*, where *Glucose—*fasting glucose in mg/dL, *Insulin*—fasting insulin (µU/mL), *NEFA*—non-esterified fatty acids (mmol/L);

*AdipoIR = Insulin × NEFA*, where *Insulin—*fasting insulin (µU/mL), *NEFA*—non-esterified fatty acids (mmol/L);

*HOMA-IR = (Glucose × Insulin)/22.5*, where *Glucose*—fasting glucose (mmol/L), *Insulin*—fasting insulin (µU/mL);

*QUICKI = 1/[log(Glucose) + log(Insulin)]*, where *Glucose—*fasting blood glucose in mg/dL, *Insulin*—fasting plasma insulin (µU/mL).

### 2.3. Statistical Analysis

Statistical analysis included the use of repeated measures ANOVA to examine the effects of treatment, week, and the treatment × week interaction on the values of blood parameters. Results are presented in the table. In the second step, linear correlation and regression analyses were performed between the independent variables (TNF-α, IL-1β, Haptoglobin, and Fibrinogen) and the dependent variables (NEFA, glucose, insulin, RQUICKI, Adipose-IR, HOMA-IR, and QUICKI). The model B_i_ = *β*_0_ + *β*_1_A*_i_*+ *ε_i_* was used, where B*_i_*—represents the value of the dependent variable for observation *_iii_*, A*_i_*—denotes the value of the independent variable for the same observation, *β*_0_—is the intercept, corresponding to the initial value of B when A equals zero, *β*_1_ is the slope, representing the effect of changes in A on B, and *ε_i_* is the residual. For clearer insight into trends during time and correlation between parameters, results were presented both graphically and in formula form. SPSS statistics software version 25 (IBM, Armonk, New York, US) was used. All statistical tests were considered significant if *p* < 0.05.

## 3. Results

The values of inflammatory parameters and IR indices, as well as the effects of the experimental treatments, are presented in Table 1. Significant changes were observed during the study period in response to ketoprofen administration on inflammatory and IR parameters. TNF-α concentrations in the control group gradually increased from a baseline value of 0.38 ± 0.05 ng/mL to 0.69 ± 0.06 ng/mL by week 2, whereas in the ketoprofen-treated group, a continuous decrease was observed from 0.35 ± 0.05 to 0.21 ± 0.05 ng/mL, with significant effects of week, treatment, and their interaction (*p* < 0.05 and *p* < 0.01, respectively). In the control group, IL-1β values increased from approximately 0.38 ng/mL at week 0 to around 0.45 ng/mL in week 1 (*p* < 0.05), before returning to near baseline levels (~0.37 ng/mL), suggesting spontaneous modulation of inflammatory activity in the absence of therapeutic intervention. In the ketoprofen-treated group, IL-1β levels remained lower throughout the period. Baseline values at week 0 were similar to those of controls (~0.36 ng/mL), but unlike controls, concentrations did not significantly increase during week 1 and significantly decreased to ~0.29 ng/mL by week 2 (*p* < 0.05). Statistical analysis indicated significant effects of treatment and week (*p* < 0.05), whereas the treatment × week interaction was not significant (NS). Haptoglobin levels in the control group increased significantly (0.41 ± 0.12 to 0.90 ± 0.09 g/L), whereas a decrease was observed in the ketoprofen-treated group (0.36 ± 0.08 to 0.24 ± 0.09 g/L), with significant effects of week, treatment, and their interaction (*p* < 0.01). Fibrinogen levels increased in the control group (6.61 ± 1.22 to 9.86 ± 1.61 g/L), while the treated group exhibited lower values (5.50 ± 1.22 to 5.08 ± 1.18 g/L), with significant differences between treatment and time (*p* < 0.05).

NEFA levels slightly decreased in the control group, whereas the ketoprofen group showed a pronounced reduction (0.92 ± 0.10 to 0.46 ± 0.07 mmol/L; *p* < 0.01). Glucose concentrations in the control group increased in week 2 (2.55 ± 0.21 mmol/L), while values in the treated group remained similar, showing a significant effect of time but not treatment (*p* < 0.05). Insulin levels decreased in both groups over the study period (*p* < 0.05), but remained more stable and slightly higher in the ketoprofen-treated group, without reaching statistical significance. The RQUICKI index increased in both groups, with higher values in the treated group (0.49 ± 0.01), indicating improved insulin sensitivity under ketoprofen administration (*p* < 0.01). Adipo-IR values showed a significant decline in both groups, particularly in the treated group (4.99 ± 0.53 to 2.35 ± 0.41; *p* < 0.01), further confirming the positive effect of the drug on metabolic balance. HOMA-IR decreased in the control group but slightly increased in the treated group during week 2, with significant effects of time and interaction (*p* < 0.05). QUICKI showed no significant changes or effects of the experimental factors. Overall, the results indicate that ketoprofen significantly reduces inflammatory markers and improves insulin sensitivity parameters during the experimental period, particularly adipose tissue insulin sensitivity.

In the next step, we examined the linear relationships between inflammatory markers and indicators of IR. All inflammatory markers showed positive correlations with NEFA (R^2^ = 0.09 to 0.26, *p* < 0.01) (Figure 1). TNF-α and fibrinogen exhibited non-significant positive correlations, whereas IL-1β and haptoglobin displayed statistically significant negative correlations with glucose (R^2^ = 0.18 and 0.24, *p* < 0.01, respectively) (Figure 2). TNF-α and fibrinogen demonstrated significant negative correlations with insulin (R^2^ = 0.24 and 0.15, *p* < 0.01, respectively), while the other two inflammatory parameters did not show statistically significant linear associations with this hormone (Figure 3). TNF-α exhibited a statistically significant negative correlation with the RQUICKI index (R^2^ = 0.13, *p* < 0.01), whereas the other inflammatory markers showed the same trend but did not reach statistical significance (Figure 4). TNF-α and IL-1β were significantly positively correlated with Adipo-IR (R^2^ = 0.05, *p* < 0.05 and R^2^ = 0.08, *p* < 0.01, respectively), whereas acute-phase proteins did not show a linear relationship with this adipose tissue IR marker (Figure 5). All inflammatory markers were negatively correlated with the HOMA-IR index (R^2^ = 0.06 to 0.17) (Figure 6). TNF-α and fibrinogen did not show statistically significant associations, while IL-1β and haptoglobin demonstrated positive correlations with the QUICKI (R^2^ = 0.13 and 0.17, *p* < 0.01, respectively) index (Figure 7).

## 4. Discussion

Concentrations of pro-inflammatory cytokines and acute-phase proteins in the control group of cows increased during the first two weeks postpartum. TNF-α and acute-phase proteins showed a sustained elevation, whereas IL-1β peaked during the first week after parturition. These concentration patterns are consistent with previous studies [24,25,26]. These cytokines stimulate immune cells and hepatocytes to synthesize acute-phase proteins, which play protective and reparative roles [27]. Therefore, measuring their concentrations is important for assessing the immune responsiveness of cows in the early postpartum period. A significant increase in haptoglobin was observed compared to later lactation stages, aligning with earlier reports [28,29]. Haptoglobin is a dynamic marker, with consistently higher values in cows with inflammatory conditions than in healthy animals [30]. Fibrinogen concentrations remained within established reference ranges [31] and were stable in clinically healthy cows during the first two weeks of lactation [32], which agrees with our results. Inflammatory processes represent part of the metabolic adaptation during early lactation and occur in clinically healthy individuals [33]. They are initiated by activation of intracellular signaling pathways and the release of TNF-α, IL-1β, and IL-6 [34]. In this study, the observed changes in TNF-α and IL-1β concentrations indicate the presence of systemic inflammation.

Administration of ketoprofen resulted in reductions in all measured inflammatory parameters, with their temporal changes exhibiting patterns opposite to those observed in the control group. The anti-inflammatory effects of ketoprofen and flunixin meglumine are associated with the inhibition of cytokine synthesis and reversible blockade of COX enzyme isoforms, predominantly COX-1 [35]. Furthermore, previous studies have reported decreased plasma TNF-α concentrations following administration of salicylates during the early postpartum period [36]. Additionally, lower haptoglobin concentrations have been observed in cows treated with lysine-acetylsalicylate [37]. Ketoprofen has also been shown to reduce TNF-α production in the mammary gland following lipopolysaccharide stimulation, further confirming its anti-inflammatory properties [38].

During early lactation, it is common to observe increased NEFA levels alongside decreases in glucose and insulin, reflecting NEB and the redirection of glucose to the mammary gland [39,40,41]. This pattern was confirmed in the present study. The insulin, glucose and NEFA concentrations were comparable to previously published results from cows in similar geographic regions [2,4,5,12,21], with no extreme values were detected within the population. Analysis of IR indices indicated the development of adipose tissue IR, as evidenced by decreased RQUICKI and increased Adipo-IR values. The HOMA-IR index showed a reduction, which could paradoxically suggest higher insulin sensitivity, while QUICKI did not change significantly over the course of the study. RQUICKI, a frequently used indicator of metabolic adaptation [42], exhibited values consistent with prior research. Although Adipo-IR has not been widely applied in cows previously, it proved useful for assessing adipose tissue IR and is a well-validated surrogate index [23]. HOMA-IR offers a simple and validated method based on the assumption of feedback regulation between the liver and pancreatic β-cells [43]. Glucose concentration is regulated through insulin-dependent hepatic glucose production, while insulin levels depend on β-cell responsiveness to glucose. In early lactation, this relationship is disrupted due to reduced feed intake and glucose redirection toward the mammary gland [44,45], so the decline in HOMA-IR represents a numerical artifact arising from the combination of low glucose and insulin values. The decreases in glucose and insulin also reflect reduced feed intake typical of early lactation, i.e., the period during which pro-inflammatory cytokines exert central effects that contribute to decreased feed consumption [46].

Ketoprofen administration resulted in reduced NEFA levels and had no effect on glucose concentrations, while insulin was non-significantly higher in treated cows during weeks 1 and 2 compared to the control group. The RQUICKI index was significantly higher, and Adipo-IR lower, indicating improved adipose tissue insulin sensitivity. HOMA-IR and QUICKI did not show significant changes. Previous studies have confirmed that NSAIDs can reduce fat mobilization [47] and inhibit lipolysis in adipocytes [48]. However, the effects of NSAIDs on lipid metabolism are not always consistent and depend on the type of drug, dose, and metabolic status of the animal. Lower NEFA concentrations following sodium salicylate administration have been reported in some studies [36], whereas other authors observed only a trend toward reduced lipolysis without statistical significance [49]. It has also been shown that salicylates can decrease circulating NEFA, particularly in multiparous cows [50], whereas selective COX-2 inhibitors, such as meloxicam, do not consistently affect lipid metabolism in healthy cows [41].

The effects of NSAIDs on glucose and insulin concentrations vary depending on the type of drug, dosage, and duration of treatment. In some studies, administration of sodium salicylate during the first days postpartum led to reductions in glucose and insulin levels, accompanied by an increase in the RQUICKI index of insulin sensitivity [36]. Other studies reported no significant changes [49], and selective COX-2 inhibitors had only minimal effects [41]. The most pronounced changes were observed during the first week of lactation, when salicylates increased peripheral insulin sensitivity, whereas other NSAIDs generally did not induce significant alterations [36,41,49]. The mechanism of action of these drugs is associated with inhibition of COX and NF-κB pathways, leading to reduced concentrations of pro-inflammatory cytokines that negatively impact insulin signaling, as well as potential alterations in hepatic gluconeogenesis [41,49].

In this study, inflammatory parameters showed positive correlations with NEFA levels and negative correlations with glucose, insulin, and insulin sensitivity indices (RQUICKI, Adipo-IR). This pattern indicates that inflammation influences the redirection of metabolic fluxes during early lactation [51]. Systemic inflammation in the peripartum period results from changes occurring before and after calving, including remodeling of the mammary gland, uterus, and digestive organs, NEB, lipolysis, and oxidative stress [52]. Elevated NEFA concentrations arising from NEB trigger so-called meta-inflammation. Saturated fatty acids, particularly palmitic acid, directly activate the TLR4 signaling pathway and the NLRP3 inflammasome [53]. Increased β-oxidation and lipid accumulation generate reactive oxygen species (ROS) and activate endoplasmic reticulum stress, which enhances NF-κB signaling and leads to apoptosis and the release of damage-associated molecular patterns (DAMPs), thereby establishing a vicious cycle between lipotoxicity and inflammation [54]. During ketosis induced by enhanced lipid mobilization, TNF-α, IL-1β, and haptoglobin concentrations increase [55], whereas administration of niacin, which has an antilipolytic effect, attenuates the inflammatory response [56]. Haptoglobin, in addition to its anti-inflammatory role, is involved in the regulation of lipid metabolism [57], which may explain its association with lipomobilization.

Inflammatory markers negatively correlate with RQUICKI and positively with Adipo-IR, confirming the association between inflammation and IR. The mechanisms linking these processes include activation of serine kinases by pro-inflammatory cytokines (TNF-α, IL-6, IL-1β), which phosphorylate IRS and reduce PI3K/AKT signaling, thereby limiting glucose uptake and decreasing insulin sensitivity. In addition, NEFA and bioactive lipids such as ceramides directly inhibit insulin signaling and activate the TLR4/NF-κB pathway in the liver and adipocytes, further enhancing the inflammatory response and acute-phase protein synthesis [58]. These processes fully support the results and correlations observed in the present study.

## 5. Conclusions

In early-lactation cows, systemic inflammation is associated with increased lipid mobilization and IR. The present results indicate that ketoprofen effectively modulates the inflammatory response, contributing to improved insulin sensitivity. Reductions in pro-inflammatory marker concentrations were accompanied by favorable changes in metabolic parameters and IR indices. Correlation analysis confirms a close association between inflammation and metabolic disturbances, with more pronounced inflammatory processes linked to decreased insulin sensitivity and increased adipose tissue IR. These findings suggest that pharmacological modulation of inflammatory activity in early-lactation cows may represent an effective approach for preventing and mitigating IR. The use of inflammatory cytokines and adipose tissue IR indices provides reliable parameters for monitoring the interplay between inflammation and IR, as well as the effects of NSAID administration, in cows during early lactation.

## Figures and Tables

**Figure 1 metabolites-15-00751-f001:**
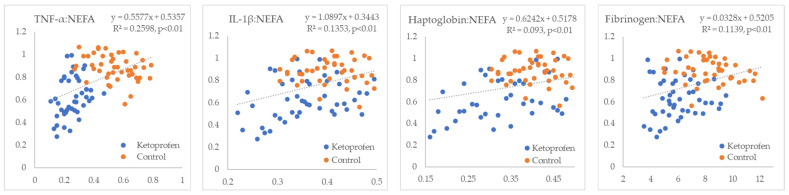
Linear regression and correlation between blood inflammatory parameters (X-axis) and NEFA (Y-axis) in cows during early lactation.

**Figure 2 metabolites-15-00751-f002:**
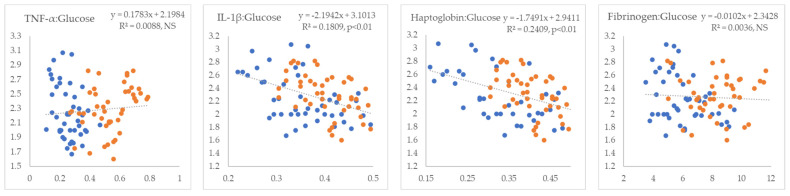
Linear regression and correlation between blood inflammatory parameters (X-axis) and glucose (Y-axis) in cows during early lactation (legend of the circle colors in Figure 1).

**Figure 3 metabolites-15-00751-f003:**
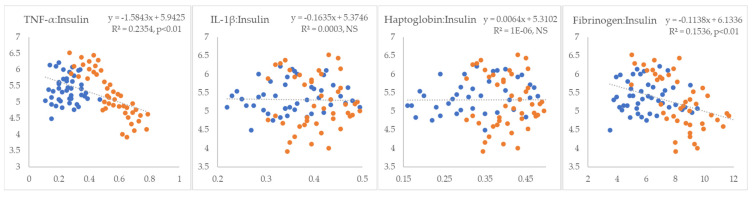
Linear regression and correlation between blood inflammatory parameters (X-axis) and insulin (Y-axis) in cows during early lactation (legend of the circle colors in Figure 1).

**Figure 4 metabolites-15-00751-f004:**
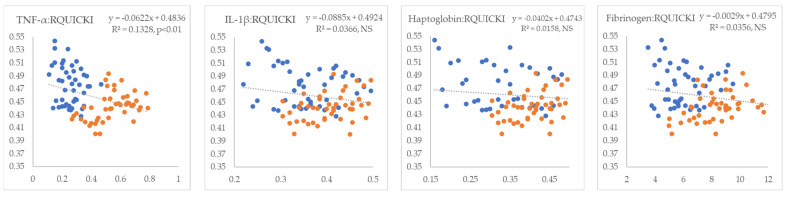
Linear regression and correlation between blood inflammatory parameters (X-axis) and RQUICKI index of IR (Y-axis) in cows during early lactation (legend of the circle colors in Figure 1).

**Figure 5 metabolites-15-00751-f005:**
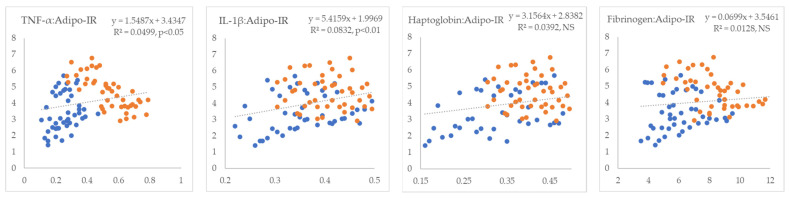
Linear regression and correlation between blood inflammatory parameters (X-axis) and Adipo-IR index of IR (Y-axis) in cows during early lactation (legend of the circle colors in Figure 1).

**Figure 6 metabolites-15-00751-f006:**
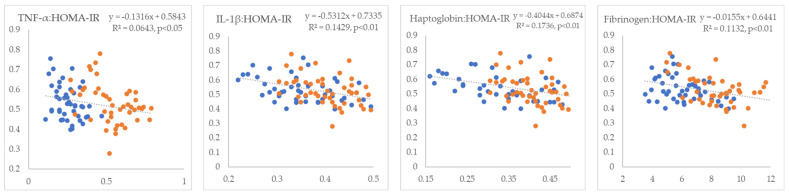
Linear regression and correlation between blood inflammatory parameters (X-axis) and HOMA-IR index of IR (Y-axis) in cows during early lactation (legend of the circle colors in Figure 1).

**Figure 7 metabolites-15-00751-f007:**
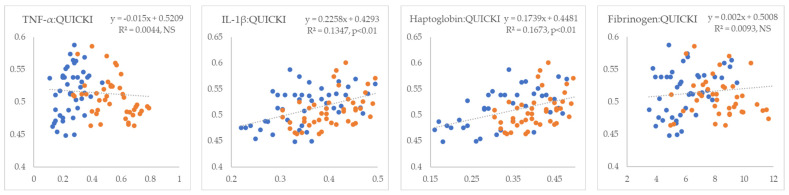
Linear regression and correlation between blood inflammatory parameters (X-axis) and QUICKI index of IR (Y-axis) in cows during early lactation (legend of the circle colors in Figure 1).

**Table 1 metabolites-15-00751-t001:** Influence of week after calving, treatment and treatment × week interaction on inflammatory and IR parameters.

Blood Parameters	Control			Ketoprofen			Treatment	Week	Treatment × Week
	Week 0	Week 1	Week 2	Week 0	Week 1	Week 2			
TNF-α (ng/mL)	0.38 ± 0.05 ^a^	0.54 ± 0.04 ^b^	0.69 ± 0.06 ^c^	0.35 ± 0.05 ^a^	0.31 ± 0.06 ^a^	0.21 ± 0.05 ^d^	<0.01	<0.01	<0.05
IL-1β (ng/mL)	0.38 ± 0.04 ^a^	0.45 ± 0.05 ^b^	0.37 ± 0.04 ^a^	0.36 ± 0.04 ^a^	0.39 ± 0.04 ^a^	0.29 ± 0.03 ^c^	<0.05	<0.05	NS
Haptoglobin (g/L)	0.41 ± 0.12 ^a^	0.84 ± 0.11 ^b^	0.90 ± 0.09 ^c^	0.36 ± 0.08 ^a^	0.49 ± 0.8 ^a^	0.24 ± 0.09 ^d^	<0.01	<0.01	<0.01
Fibrinogen (g/L)	6.61 ± 1.22 ^a^	8.69 ± 1.51 ^b^	9.86 ± 1.61 ^b^	5.50 ± 1.22 ^a^	7.53 ± 1.25 ^c^	5.08 ± 1.18 ^a^	<0.05	<0.05	<0.05
NEFA (mmol/L)	0.94 ± 0.11 ^a^	0.82 ± 0.09 ^b^	0.73 ± 0.09 ^b^	0.92 ± 0.1 ^a^	0.61 ± 0.09 ^c^	0.51 ± 0.7 ^d^	<0.01	<0.01	<0.01
Glucose (mmol/L)	2.29 ± 0.26 ^a^	2.00 ± 0.25 ^a^	2.55 ± 0.21 ^b^	2.16 ± 0.23 ^a^	2.04 ± 0.25 ^a^	2.64 ± 0.26 ^b^	NS	<0.05	NS
Insulin (mU/L)	6.10 ± 0.51 ^a^	5.14 ± 0.43 ^b^	4.50 ± 0.48 ^c^	5.83 ± 0.41 ^a^	5.23 ± 0.43 ^b^	5.09 ± 0.55 ^b^	NS	<0.05	NS
RQUICKI	0.48 ± 0.02 ^a^	0.46 ± 0.01 ^b^	0.45 ± 0.01 ^b^	0.45 ± 0.01 ^b^	0.49 ± 0.02 ^c^	0.49 ± 0.01 ^c^	<0.01	<0.01	<0.05
Adipo-IR	5.51 ± 0.52 ^a^	4.40 ± 0.59 ^b^	3.81 ± 0.54 ^c^	4.99 ± 0.53 ^a^	3.17 ± 0.48 ^d^	2.35 ± 0.41 ^e^	<0.01	<0.01	<0.01
HOMA-IR	0.61 ± 0.08 ^a^	0.46 ± 0.07 ^b^	0.51 ± 0.09 ^b^	0.57 ± 0.08 ^a^	0.47 ± 0.07 ^b^	0.60 ± 0.08 ^a^	NS	<0.05	NS
QUICKI	0.55 ± 0.03 ^a^	0.53 ± 0.02 ^a^	0.52 ± 0.03 ^a^	0.53 ± 0.03 ^a^	0.55 ± 0.03 ^a^	0.54 ± 0.02 ^a^	NS	NS	NS

^a,b,c,d,e^ Different superscripts mean significant differences between values.

## Data Availability

Original data can be available after personal communication with first author.

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
