# Peer review of "Relation Between Inflammatory Parameters and Insulin Resistance Indices in Cows During Early Lactation"

_metabolites, 2025, doi:10.3390/metabo15110751_

Round 1
Reviewer 1 Report
Comments and Suggestions for Authors
This manuscript presents a well-designed and timely study investigating the interplay between systemic inflammation and insulin resistance (IR) in early-lactation dairy cows, with a focus on the effects of ketoprofen administration. The topic is of significant relevance to dairy science and veterinary medicine, given the metabolic challenges faced by high-yielding cows during the transition period. The study is methodologically sound, with appropriate animal grouping, sampling timeline, and statistical analyses. The inclusion of multiple inflammatory markers and IR indices strengthens the comprehensiveness of the work.
Minor Weaknesses and Suggestions for Improvement:
Minor Weaknesses and Suggestions for Improvement:
-
Introduction:
-
The introduction could be strengthened by briefly explaining why adipose tissue IR is particularly relevant in early lactation, and how it differs from systemic IR in this context.
-
Consider adding a sentence or two on why ketoprofen was chosen over other NSAIDs, and its known effects in ruminants.
-
-
Methods:
-
Please clarify the exact timing of blood sampling relative to calving (e.g., day 0 = day of calving?). This would improve reproducibility.
-
The formula used for calculating IR indices (RQUICKI, Adipo-IR, etc.) should be explicitly stated in the methods or referenced clearly.
-
-
Results:
-
The non-significant changes in QUICKI and the paradoxical decrease in HOMA-IR in controls deserve a more detailed explanation in the results or discussion.
-
Consider presenting correlation coefficients (R²) and p-values directly in the text for key relationships, not only in figures.
-
-
Discussion:
-
The discussion would benefit from a deeper exploration of why ketoprofen did not significantly affect glucose or insulin levels, yet improved RQUICKI and Adipo-IR.
-
It would be valuable to discuss the potential practical implications of using NSAIDs like ketoprofen as a metabolic management tool in early lactation.
-
-
Language and Clarity:
-
The manuscript is generally well-written, but some sentences are lengthy and could be simplified for better readability.
-
Ensure consistency in abbreviations (e.g., IR is defined twice).
-
Author Response
Response
This manuscript presents a well-designed and timely study investigating the interplay between systemic inflammation and insulin resistance (IR) in early-lactation dairy cows, with a focus on the effects of ketoprofen administration. The topic is of significant relevance to dairy science and veterinary medicine, given the metabolic challenges faced by high-yielding cows during the transition period. The study is methodologically sound, with appropriate animal grouping, sampling timeline, and statistical analyses. The inclusion of multiple inflammatory markers and IR indices strengthens the comprehensiveness of the work.
AUTHORS: Thank you very much for your support.
Minor Weaknesses and Suggestions for Improvement:
- Introduction:
- The introduction could be strengthened by briefly explaining why adipose tissue IR is particularly relevant in early lactation, and how it differs from systemic IR in this context.
- Consider adding a sentence or two on why ketoprofen was chosen over other NSAIDs, and its known effects in ruminants.
AUTHORS: Thank you. Added.
- Methods:
- Please clarify the exact timing of blood sampling relative to calving (e.g., day 0 = day of calving?). This would improve reproducibility.
- The formula used for calculating IR indices (RQUICKI, Adipo-IR, etc.) should be explicitly stated in the methods or referenced clearly.
AUTHORS: Thank you. Added.
- Results:
- The non-significant changes in QUICKI and the paradoxical decrease in HOMA-IR in controls deserve a more detailed explanation in the results or discussion.
- Consider presenting correlation coefficients (R²) and p-values directly in the text for key relationships, not only in figures.
AUTHORS: Thank you. Added.
- Discussion:
- The discussion would benefit from a deeper exploration of why ketoprofen did not significantly affect glucose or insulin levels, yet improved RQUICKI and Adipo-IR.
- It would be valuable to discuss the potential practical implications of using NSAIDs like ketoprofen as a metabolic management tool in early lactation.
AUTHORS: Explanation exists in this section. Line 295-309 in v2.
- Language and Clarity:
- The manuscript is generally well-written, but some sentences are lengthy and could be simplified for better readability.
- Ensure consistency in abbreviations (e.g., IR is defined twice).
AUTHORS: Thank you. MDPI English editors will help…
Reviewer 2 Report
Comments and Suggestions for Authors
In this manuscript, the authors examined the relationship between inflammatory parameters and insulin resistance indices in cows during early lactation. The primary objective of the study is to investigate the effect of ketoprofen on inflammatory parameters and IR indices, as well as to thoroughly evaluate the relationship between surrogate IR indices (HOMA-IR, QUICKI, RQUICKI, Adipo-IR) and a panel of key inflammatory markers, including cytokines (TNF-α, IL-1β) and acute phase proteins (haptoglobin, fibrinogen). It is hypothesized that ketoprofen administration will significantly alter the concentrations of inflammatory markers, carbohydrate and lipid metabolism indices, and IR indices, and that these correlations will reflect the interaction between inflammation and IR. IR, defined as the reduced ability of target tissues (muscle, liver, and adipose tissue) to effectively respond to circulating insulin, is an important adaptation during early lactation, allowing glucose to be conserved and redirected to the mammary gland. Given the close relationship between inflammation and IR, there is significant interest in therapeutic strategies aimed at reducing the inflammatory burden. It is believed that reducing inflammation can improve insulin signaling and thereby mitigate the pathological manifestations of IR. However, comprehensive studies that simultaneously integrate a broader panel of IR indices and inflammatory markers in the context of ketoprofen use in dairy cows during early lactation are lacking. This demonstrates the originality and relevance of this research topic. These results will expand scientific knowledge and demonstrate that inflammatory cytokines and adipose tissue IR indices serve as reliable parameters for monitoring the interaction between inflammation and IR, as well as for assessing the metabolic effects of NSAID treatment in cows in early lactation. The authors fully achieved the stated goal and objectives. The experimental portion of the study was conducted to a high standard, adhering to modern research methods. The materials and methods were selected in accordance with the aim and objectives of the manuscript. The study results were statistically processed, allowing for a comprehensive assessment and analysis. The manuscript is logically structured with a clear text structure. The manuscript is written in understandable English. The authors competently use specialized terms and professional vocabulary. The text contains some typos and inaccuracies (spelling and typographical). The authors conducted a comparative analysis of the obtained data. The obtained data were compared with existing literature. The data presented in the manuscript are adequately analyzed. The authors' conclusions are consistent with the research results. The arguments for the conclusion follow naturally from the material. The conclusion provides answers to the aim and objectives of the study. The reference list fully allows for an analysis of the literature related to the research area covered in the manuscript. The list of references is sufficient in terms of content. The authors cite the literature accurately in the manuscript. Incorrect citations are not permitted. However, the following comments are possible: 1. Materials and Methods. There is no information about the study being reviewed by an ethics committee. 2. Materials and Methods. Section 2.2. The main indicators (indices) that were calculated should be described. 3. Figures. Axes and units of measurement are missing. 4. Conclusion. It should be more specific and described more precisely. Correcting these comments will improve the quality of the manuscript and increase reader interest. I recommend revising the manuscript.
Author Response
Thank you for your review, support, and encouraging comments. We will make the necessary corrections, both independently and with the assistance of MDPI technical editors. From the very beginning, we considered whether it was necessary to specifically label the X and Y axes, and we did not do so because a linear equation is already provided, which implies which axis represents X and which represents Y, and there are descriptions in the text below the figures. We will consult with MDPI technical editors, and if deemed necessary, we will label the graphs accordingly. The ethical approval number has been included, but the date was not, so we have now provided the specific date from the ethical approval.
Reviewer 3 Report
Comments and Suggestions for Authors
This study offers valuable insights into how systemic inflammation, lipid mobilization, and insulin resistance interact in cows during early lactation. The results show that ketoprofen can effectively reduce inflammation, which in turn improves insulin sensitivity and overall metabolic balance. Using inflammatory cytokines and adipose tissue insulin-resistance markers as monitoring tools also seems to be a promising approach for tracking the link between inflammation and metabolic disorders.
Looking ahead, it would be interesting to investigate how long-term NSAID use might affect cow health and productivity, and whether there are differences between breeds or management systems. Overall, the authors should be congratulated for a well-designed and comprehensive study that makes a meaningful contribution to understand of how inflammation can be managed in dairy cows during the early stages of lactation.
Author Response
Thank you for your support and for your truly kind words. The long-term effects of NSAIDs could be a topic for future investigation. This is particularly interesting in the context of ketoprofen use, as it rapidly reaches its maximum effect and is then quickly eliminated from the body. Metabolic stresses during peak lactation and other metabolic disorders could potentially be modulated by the prolonged use of this NSAID. We appreciate your comprehensive perspective on this topic.